# Addressing Body Image Disturbance through Metaverse-Related Technologies: A Systematic Review

**Moises Perez [1], Adriana Pineda-Rafols [2,3], Maria Pilar Egea-Romero [2,3], Maria Gonzalez-Moreno [2,4]** and **Esther Rincon [2,3,*]**

[1]  Instituto Superior de Estudios Psicológicos (ISEP), 28010 Madrid, Spain; moisesperezpsicologia@gmail.com

[2]  Psycho-Technology Lab, Universidad San Pablo-CEU, CEU Universities, Urbanización Montepríncipe, 28660 Boadilla del Monte, Spain; adriana.pinedarafols@ceu.es (A.P.-R.); pegea@ceu.es (M.P.E.-R.); mgmoreno@ceu.es (M.G.-M.)

[3]  Departamento de Psicología y Pedagogía, Facultad de Medicina, Universidad San Pablo-CEU, CEU Universities, Urbanización Montepríncipe, 28660 Boadilla del Monte, Spain

[4]  Departamento de Ciencias Médicas Básicas, Facultad de Medicina, Universidad San Pablo-CEU, CEU Universities, Campus de Montepríncipe, Urbanización Montepríncipe, 28660 Boadilla del Monte, Spain

*  Correspondence: maria.rinconfernande@ceu.es

**Abstract:** There is an increasing awareness about body image disturbance and eating disorders which calls for a multidisciplinary approach. The usefulness of new technologies for improving body image distortion has been addressed in the scientific literature, but has not included current strategies such as Metaverse-related technologies. Seemingly, this is the first systematic review which focuses on the efficacy of Metaverse-related technologies in reducing body image disturbance related to eating disorders like *Anorexia nervosa* and *Bulimia nervosa*. The main objective of this study was to review the scientific studies published in the last decade to answer the following three questions: (1) Are Metaverse-related technologies useful in mitigating body image disturbance in patients diagnosed with *Anorexia nervosa* and *Bulimia nervosa*? (2) What are the advantages and disadvantages? (3) Are the patients satisfied after using this kind of technology? The results obtained were that 80% of the included studies found metaverse-related technologies useful for the improvement in body image disturbance, in addition to various advantages, such as a decrease in eating disorder symptomatology. Whereas patient satisfaction was only evaluated in 20% of the included studies, with the majority of patients evaluating the use of metaverse-related technologies positively. We will conduct a systematic review of the peer-reviewed literature from EBSCO Discovery Service, and Web of Science (WOS), following the PRISMA statements. Only Journal articles published from 2013 to the present, written in the English language, will be reviewed. The findings are expected to offer valuable insights for the development of novel approaches for the improvement in body image disturbance in healthcare settings.

**Keywords:** metaverse; mixed reality; extended reality; virtual reality; augmented reality; body image; body image distortion; body image disturbance; body size estimation; eating disorders; *Anorexia nervosa*; *Bulimia nervosa*

## 1. Introduction

An eating disorder (ED) is a severe and specific alteration in the consumption of food that is observed in people who have distorted patterns when eating, who are characterized by eating excessive amounts of food or not eating at all, which is caused by a psychological impulse instead of a biological or metabolic need [1]. Eating disorders (ED) are more common in women and generally appear during adolescence or early youth, although it can also appear in childhood or as an adult [2].

The two main types of ED are *Anorexia nervosa* (AN) and *Bulimia nervosa* (BN) [1]. According to the American Psychiatric Association (2013) [3] *Anorexia nervosa* is the "re-

striction of energy intake leading to a significantly low body weight in the context of age, sex, developmental trajectory, and physical health". It also includes an immense fear of gaining weight, even when at a very low body weight, accompanied by behavior to lose weight and a negative body image [3]. *Bulimia nervosa* (BN) includes recurring episodes of binge-eating (a sense of lack of control when eating an excessive amount of food, within a 2 h period) and "recurrent inappropriate compensatory behaviors in order to prevent weight gain", such as vomiting, the use of laxatives, exercise, fasting, etc. [3]. AN and BN are two types of ED that alter emotional and cognitive processes that can deteriorate and affect one's quality of life [4].

In 2017, a systematic review by Lindvall Dahlgren et al. [5] included data on the prevalence of AN and BN including woman and men of all ages, from studies conducted in various countries (Canada, The United Kingdom, USA, Sweden, Germany, Finland, Australia, Portugal Switzerland and the Netherlands). It was observed that the prevalence for AN in women was between 0.67–1.2% while in men it was 0.1%. However, for BN, the prevalence in woman was 0.62%. In a meta-analysis by Qian et al. [6], which pooled data from people aged 15 years and older worldwide from 1984 to 2017, they found that the prevalence of BN in men was 0.4%. In a systematic review and meta-analysis of 31 studies, published between 1980 and 2019, the incidence rates found in Spanish woman aged 12–22 years old suffering from AN was of 200 cases for 100,000 people a year [7].

Body image disturbance is considered to be one of the most prevalent clinical characteristics in eating disorders [8–11] and is also a relevant prognosis factor for someone suffering from *Anorexia nervosa* [12] or *Bulimia nervosa* [12–14]. As stated by Cash and Pruzinsky [15], body image is the multidimensional perception (cognitive, behavioral, affective and evaluative components) of one's body. Yamamotova et al. [16] divided body image between four different components: cognitive (thoughts and beliefs of one's body); perceptual (how one perceives the size and form of their body and each part); affective (one's feelings of their body); and behavioral (actions to verify, alter or hide one's body). Therefore, one's negative feelings and thoughts about their own body is considered to be body image disturbance [17].

Body image disturbance is usually segmented into two distinct and separate categories [18,19], the first being body size distortion, which consists of a distorted perception of one's body; be it the perceptual distortion of the whole body or specific parts of the body [20]. Meanwhile, the second type of body image distortion is defined as body dissatisfaction, which includes negative cognitive, affective or attitudinal thoughts towards one's body image [20]. Keeton et al. [19] described these two types of body image disturbance as independent and distinct. Body image disturbance is associated with worrisome food habits, such as restrictive diets, a lack of control when eating and even bulimic tendencies [14]. Consequently, improving one's body image is pivotal in the management of an eating disorder, to such a degree that the persistence of a distortion in how one sees their body after treatment is a dependable indicator of a relapse in anorexia or bulimia [14]. Even though body image disturbance is a frequent factor in people suffering from AN or BN, it is not exclusive to them, if not also experienced in different degrees by people without eating disorders [21]. Some of those who suffer from body image disturbance outside of an ED can still find it to be disabling and can even be associated with depression [22].

Many studies [11,21–23] indicate that interventions targeting the improvement in body image distortion could be helpful for treatments for ED. Along with the increase in AN and BN these past years [24], there has also been an increase in the use of metaverse-related technologies in healthcare environments, with the use of virtual reality being the most prominent [25].

*Metaverse-Related Technologies Use in ED*

The Metaverse is a "next-generation internet hypothesis consisting in a stable, decentralized, 3D virtual reality setting" [26]. It is also understood as "a digital universe accessible through a virtual setting" [27] which links "the physical world with a virtual

environment" [28]. Metaverse technology includes four categories: augmented reality, lifelogging, virtual reality, and the mirror world [29]. The four categories of Metaverse are divided by two axes: augmentation versus simulation and external versus intimate [28]. Augmentation technology is that of which is added to the existing environment (the real world); thus, it imposes digital elements on the physical world [28]. Consequently, augmented reality (AR) combines both the digital and the real world by using digital technologies, such as sounds, visual elements, or other perceptual experiences to enhance the real world [30]. Alternatively, simulation technology creates virtual interactions and experiences [27]; therefore, virtual reality (VR) completely replaces the actual physical world with a digitally generated virtual environment [31]. According to Al-Rasheed [32], VR is a computer-based interface, which generates a virtual environment that allows people to interact with it as if they were in another world.

Lifelogging consists of technology that records everyday information of individuals by using AR technology (ex: Facebook or Instagram) whereas mirror worlds build virtual maps and models that are representations of real-world places and environments, by using GPS (global positioning system) technology (ex: Google Earth or Google Maps), to mirror a physical context into an online database [28].

The difference between the external and the intimate world is that the first focuses on the environment and how to control one's surroundings, whereas the intimate world has to do with the identity and behavior of objects and different individuals by creating avatars or digital profiles that represent these identities [28].

VR appears to have a positive impact on mitigating body image distortions and increasing one's self esteem and self-efficacy [32]. VR uses equipment, such as headsets, position trackers, and other tools [33] which allows the person to feel completely immersed in that world. VR can be applied to change someone's emotions, behavior and lifestyle, therefore making it understandable to use in any healthcare field [34], and can also simulate virtual scenarios, closely resembling real-world experiences [35], which enables people to confront frightening situations from a safe and controlled setting, which in turn, lessens that persons resistance towards exposure [36].

The incorporation of avatars represents one of the latest technological advancements [37–39] which are being used to immerse patients in stressful virtual situations, to better their body image perceptions and help develop healthier habits [40]. There is different software, such as "Virtual and body" [41], that help with body image distortions. "Virtual and body" contains six different virtual training environments to help with patients' perceptions of their bodies [41] using metaverse-related technologies.

Lastly, in a study with non-clinical participants, carried out by Preston and Ehrsson [42], a direct link was found between the perception of one's body and an explicit emotional experience (body satisfaction) in the female sample. These emotional changes are associated with cognitive–behavioral eating disorder characteristics, on account of people who have more thoughts and behaviors associated with ED, have a weaker affective representation of their body, which leads them to a greater variation in body satisfaction due to their perception (real or illusory) of their body size.

Due to the limited studies that define the specific efficacy of Metaverse-related technology in the improvement in body image disturbances in patients diagnosed with AN and BN, the current study pursues the following objectives: (1) to determine the efficacy of metaverse-related technologies in enhancing body image disturbance in patients diagnosed with *Anorexia nervosa* and *Bulimia nervosa*; (2) to detect the advantages and disadvantages of its use with these patients; and finally (3) to determine the degree of satisfaction reported by patients after its use.

## 2. Materials and Methods

### 2.1. General Description

In July 2023, a systematic search strategy was executed to identify all pertinent studies involving the use of Metaverse-related technologies to increase body image distortion

in eating disorders as *Anorexia nervosa* and *Bulimia nervosa*. The systematic review was conducted and reported in accordance with the Preferred Reporting Items for Systematic Reviews and Meta-Analyses (PRISMA) Statement (see study protocol in Supplementary File S1) [43]. The protocol was registered with the PROSPERO International Prospective Register of Systematic Reviews (CRD42023449337).

*2.2. Selection Criteria*

Inclusion criteria: Relevant study papers were limited to journal articles focusing on the use of Metaverse-related technologies to increase body image distortion in eating disorders such as *Anorexia nervosa* and *Bulimia nervosa*. Included studies were published in the English language, within the last decade (between 2013–July 2023), providing specific outcomes (quantitative results).

Exclusion criteria: Studies involving the use of Metaverse-related technologies but not including body image distortion in patients diagnosed with *Anorexia nervosa* or *Bulimia nervosa* were excluded. Furthermore, those manuscripts in which Metaverse-related technology was used to help other ED patients such as obesity or binge eating, or were not focused on body image distortion were discarded. Exclusions encompassed protocols with unpublished results, narrative reviews, non-journal articles (congress abstracts, book chapters or theses), and publications in languages other than English. Congress abstracts were discarded due to their limited content, which does not provide enough information to properly analyze the primary and secondary outcomes outlined in the present study.

*2.3. Outcomes*

The primary outcomes were the type of training using Metaverse-Related Technologies developed, the kind of Metaverse-Related Technology used, its usefulness to enhance or improve Body Image Disturbance in AN and BN patients, and finally, the questionnaire used for patient's assessment. The secondary outcomes assessed the primary advantages and disadvantages of the trainings developed, as well as the patients' satisfaction levels following the utilization of Metaverse-Related Technology.

*2.4. Search Methodology*

An extensive search was conducted across EBSCO Discovery Service, and Web of Science (WOS), from inception until July 2023. The detailed search strategies employed for all databases are outlined in the Supplementary File S1. All original research articles were retrieved for examination, and a search library was created using RefWorks©, a bibliography management program.

*2.5. Data Collection and Analysis*

Two authors (M.P. and E.R.) independently assessed and reviewed all titles and abstracts for completeness through a three-phase process: firstly, the titles of the records were assessed, followed by their abstracts. Finally, if a reference was deemed relevant by a reviewer, after evaluating both titles and abstracts, the full paper was retrieved. After this, an inter–rater agreement between the 2 investigators (M.P. and E.R.) was calculated using Cohen kappa scores. The interpretation of the Cohen kappa coefficient was calculated using SPSS version 27 (IBM Corp., New York, NY, USA) and was based on the categories developed by Douglas Altman [44]: 0.00–0.20 (poor), 0.21–0.40 (fair), 0.41–0.60 (moderate), 0.61–0.80 (good), and 0.81–1.00 (very good). In case of discrepancies, a third author was consulted (A.P-R.). Cross-checking procedures were executed to identify any inaccuracies or oversights (E.R.). Any other discrepancies were resolved amongst the core team with the involvement of the broader research team when necessary.

*2.6. Data Extraction and Management*

We gathered data encompassing the following categories: (1) publication year, (2) country, (3) study design, (4) study aim, (5) sample size and mean participants' age, (6) eating

disorder addressed, (7) body image disturbance tested, (8) training using Metaverse-Related Technologies, (9) Metaverse-Related Technology used, (10) useful to improve Body Image Disturbance, (11) questionnaire used, (12) main advantages/disadvantages, and (13) students' satisfaction.

*2.7. Quality of Studies Included*

Considering the variety of research designs, we assessed the quality of the included studies using the Mixed Methods Appraisal Tool (MMAT) developed in 2006 [45] and revised in 2018 [46]. The overall scores with the highest values indicated a lower quality of included studies (see in Supplementary File S1). Two authors (P.E.-R. and M.G.-M.) independently extracted data on outcomes from all studies. Data were reviewed for completeness by one reviewer (E.R.).

*2.8. Statistical Analysis*

Data were pooled using the program SPSS v. 27 (IBM Corp., New York, NY, USA), enabling an analysis of both frequencies (percentages) and mean values.

## 3. Results

*3.1. Study Selection and Inclusion*

A total of 1109 records were included in RefWorks©, through the electronic database search. After removing 552 duplicates, 557 records underwent a title and abstract assessment. Of those, 476 were excluded as they clearly did not meet the inclusion criteria. Consequently, 81 papers were deemed eligible for a full text review; 71 of them [47–115] being excluded for various reasons (see Supplementary File S3). A total of 10 publications were finally included [116–125]. The Cohen kappa coefficient demonstrated a substantial level of agreement, being categorized as "good" (κ = 0.67) (range 0.61–0.80) based on the categories developed by Altman [44]. A PRISMA flow diagram [43] is provided in Figure 1. All selected studies were considered to be of a sufficient quality to contribute equally to the thematic synthesis.

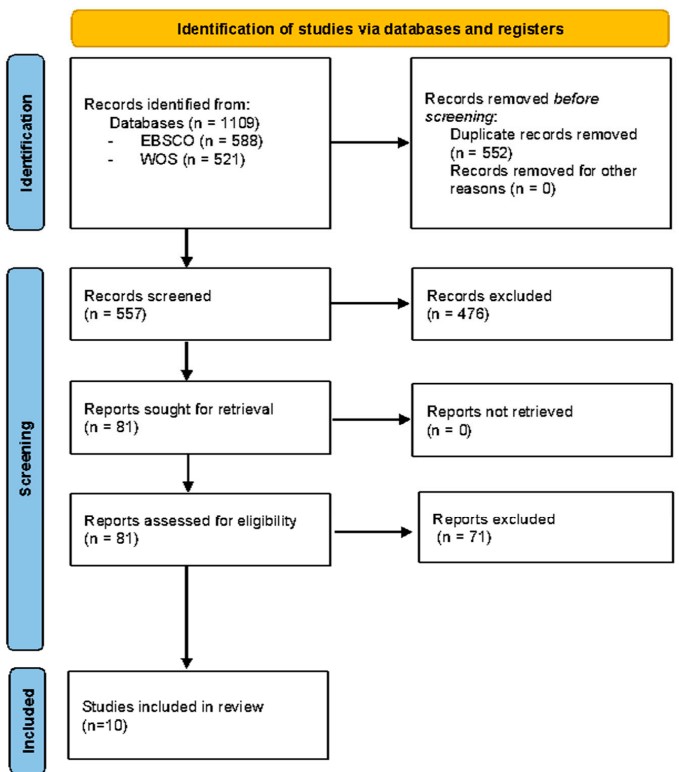

**Figure 1.** Systematic review of the literature flowchart.

### 3.2. General Characteristics of the Studies Included

Regarding points 1 (year of publication), 2 (country of the study) and 3 (study design), the following results were extracted (Table 1): the 10 selected studies were published between 2013 (*n* = 1; 10%) [121] and 2023 (*n* = 1; 10%) [49]. The majority of the studies were conducted in Spain (*n* = 5; 50%) [117,118,121–123]. The remaining papers were published in Italy (*n* = 3; 30%) [120,124,125], Germany (*n* = 1; 10%) [116] and the Netherlands (*n* = 1; 10%) [119]. All the studies involved followed a quantitative approach (*n* = 10; 100%) [116–125] (Table 1).

**Table 1.** General characteristics of included studies (*n* = 10).

| Study | Publication Year | Country | Study Design |
|---|---|---|---|
| Behrens et al. [116] | 2023 | Germany | Quantitative |
| Porras-Garcia et al. [117] | 2020 | Spain | Quantitative |
| Porras-Garcia et al. [118] | 2021 | Spain | Quantitative |
| Keizer et al. [119] | 2016 | Netherlands | Quantitative |
| Malighetti et al. [120] | 2021 | Italy | Quantitative |
| Marco et al. [121] | 2013 | Spain | Quantitative |
| Porras-Garcia et al. [122] | 2020 | Spain | Quantitative |
| Porras-Garcia et al. [123] | 2020 | Spain | Quantitative |
| Provenzano et al. [124] | 2020 | Italy | Quantitative |
| Serino et al. [125] | 2019 | Italy | Quantitative |

In regard to points 4 (study aim), 5 (sample size and mean participants' age), 6 (Eating Disorder) and, 7 (Body Image Disturbance tested), the following findings were extracted: The objectives of the studies were extremely varied (Table 2). The sample size ranged from 1 [117,123,125] to 59 participants involved [119], although some studies [*n* = 4; 40%] divided participants between two groups: the experimental (people suffering from AN or BN) and the control group [118,119,122,124]. One of the studies (10%), had two different experiments with different participants in each one, only including the mean participants age from one of the experiments [116]. Some studies (*n* = 5; 50%) did not provide the mean participant's age [116,118,119,122]. The mean age of participants ranged from 14 [123] to 30 [116] years. Al the included studies involved patients with AN (*n* = 10; 100%) [116–125], and only one included BN (*n* = 1; 10%) [121].

**Table 2.** General characteristics of included studies (II) (*n* = 10).

| Study | Study Aim | Sample Size (Mean Age) | Eating Disorder | Body Image Disturbance Tested |
|---|---|---|---|---|
| Behrens et al. [116] | To evaluate the usefulness of VR exposure to a healthy body | CG: 20(26.36) EG: 20(30.1) | EG: AN; GC: No AN | Cognitive/Perceptual/Affective |
| Porras-Garcia et al. [117] | To evaluate the effectiveness of VR body exposure therapy as adjunct treatment | 1(15) | AN | Cognitive/Perceptual/Affective |
| Porras-Garcia et al. [118] | To evaluate the effectiveness of VR body exposure therapy to reduce fear of weight gain and other ED symptoms | 35(NP) | EG: AN; GC: No AN | Cognitive/Perceptual/Affective |
| Keizer et al. [119] | To investigate whether a *Full-Body Illusion* (FBI) in VR affects body size estimation | 59(NP) | EG: AN; GC: No AN | Perceptive |
| Malighetti et al. [120] | To use VR to modify body image distortions and negative body-related memories | 7(17) | AN | Cognitive/Perceptual/Affective |
| Marco et al. [121] | To evaluate the effectiveness of cognitive behavioral therapy (CBT) supported by VR in body image treatment | 32(21.82) | AN/BN | Cognitive/Affective |

**Table 2.** *Cont.*

| Study | Study Aim | Sample Size (Mean Age) | Eating Disorder | Body Image Disturbance Tested |
|---|---|---|---|---|
| Porras-Garcia et al. [122] | To evaluate the usefulness of VR body exposure therapy in AN treatment | 17(NP) | EG:AN; GC: No AN | Perceptual/Affective |
| Porras-Garcia et al. [123] | To evaluate the usefulness of VR body exposure therapy in AN treatment | 1(14) | AN | Cognitive/Perceptual/Affective |
| Provenzano et al. [124] | To use VR to characterize and reduce body image distortion | 40(23.6) | EG:AN; GC: No AN | Affective |
| Serino et al. [125] | To report on the use of VR using *Full-Body Illusion* as part of a multidisciplinary treatment | 1(NP) | AN | Perceptual |

NP = No provided by authors. EG = Experimental Group. CG = Control Group.

### 3.3. Assessment of Methodological Quality of Included Studies

Although all included studies followed a quantitative design, extensive heterogeneity was found in the statistical methods used with diversity in the presentation of the results obtained. In many cases, the description of randomization procedures was insufficient, as well as information on main variables being manipulated (or not) by researchers (see in Supplementary File S2).

### 3.4. Primary Outcomes

Regarding points 8 (training using Metaverse-Related Technologies), 9 (Metaverse-Related Technology used), 10 (useful to improve Body Image Disturbance), and 11 (Questionnaire used), the subsequent findings were retrieved (Table 3).

**Table 3.** Primary outcomes (*n* = 10).

| Study | Training Using Metaverse-Related Technologies | Metaverse-Related Technology Used | Useful to Improve Body Image Disturbance | Questionnaire Used |
|---|---|---|---|---|
| Behrens et al. [116] | In four VR sessions, participants were exposed to a healthy BMI, measuring different aspects of Body Image Distortion before and after each session. | VR (*Valve Index—VIVE*) | No | *BCCS* y *FKB-20* [126,127] |
| Porras-Garcia et al. [117] | A virtual avatar was created, that gradually increased its BMI over five sessions, discontinuing when the patient's body anxiety decreased by 40% during a measurement 5 months later. | VR (*HMD-HTC-VIVE*) | Yes | HMD Fove Eye Tracking, *TSA-D*, PASTAS y *VAS-A* [128,129] |
| Porras-Garcia et al. [118] | Patients were evaluated before, 5 weeks after and at a 3-month follow-up. The experimental group received VR body exposure along with CBT, assessing anxiety during the sessions. | VR (*HTC-VIVE HMD*) | Yes | FOVE VR-HMD, *BAS, EDI-BD, PASTAS & BIAS-BD* [129–132] |
| Keizer et al. [119] | In a split-group trial, the VR Full-Body Illusion was applied, and the body perception and estimation were assessed before, during and after exposure to an avatar. | VR (*Oculus Rift DK2*) | Yes | Analog measurements |
| Malighetti et al. [120] | Full Body Illusion was induced with VR, assessing body image and estimates of actual and ideal size before and after exposure. | VR (NP) | Yes | *BSQ*, BSS and VR Body Size Estimation Task [133,134] |
| Marco et al. [121] | CBT was combined with VR intervention, assessing different aspects of Body Image Distortion before and after treatment, and one year later. | VR (*V6 de Virtual Research*) | Yes | *BIATQ, BAT, BASS & SIBID* [135–138] |

**Table 3.** *Cont.*

| Study | Training Using Metaverse-Related Technologies | Metaverse-Related Technology Used | Useful to Improve Body Image Disturbance | Questionnaire Used |
|---|---|---|---|---|
| Porras-Garcia et al. [122] | In a controlled trial, CBT was combined with five sessions of VR body exposure therapy, assessing multiple components of body image before and after treatment. | VR (HTC-VIVE) | Yes | *VR HMD FOVE, BIAS-BD, BIAS-X, EDI-BD, PASTAS &BIAS-O* [129,131,132] |
| Porras-Garcia et al. [123] | During five sessions, a patient was exposed to an avatar with VR that gradually increased her BMI, evaluating different aspects of body image and reducing anxiety by 40%. | VR (HMD-HTC-VIVE) | Yes | *TSA-BD, TSA-D, EDI-BD & PASTAS* [128,129,131] |
| Provenzano et al. [124] | Using VR and the Full Body Image Illusion, participants were exposed to avatars of different sizes and their emotional responses and degree of attraction to these avatars were assessed. | VR (*Oculus Rift Developers Kit Dk1*) | No | Not Provided (indirect measurements used) |
| Serino et al. [125] | The ability of VR to improve a patient's body perception was evaluated by comparing actual measurements with estimations in three sessions using different visuotactile stimulation conditions in Full Body Illusion. | VR (*HMD-Oculus Rift DK2*) | Yes | Not Provided (analog measurements used) |

VR = Virtual Reality. BCCS: Body Checking Cognitions Scale. FKB-20: Body Image Questionnaire. HMD-HTC-VIVE: PC Virtual Reality System. TSA-D: Silhouette Test for Adolescents (Body image distortion). PASTAS: Physical Appearance State and Trait Anxiety Scale. VAS-A: Visual Analog Scales (body anxiety). FOVE VR-HMD: Eye Tracking Virtual Reality headset. BAS: The Body Appreciation Scale. EDI-BD: Eating Disorder Inventory (Body Dissatisfaction). BIAS-BD: Figural Drawing Scale for Body Image Assessment. BSQ: Body Shape Questionnaire. BSS: Body Satisfaction Scale. BIATQ: Body Image Automatic Thoughts Questionnaire. BAT: Body Attitude Test. BASS: Body Areas Satisfaction Scale. SIBID: Situational Inventory of Body-Image Dysphoria. TSA-BD: Silhouette Test for Adolescents (Body Dissatisfaction). BIAS-X: Figural Drawing Scale for Body Image Assessment (Body distortion). BIAS-O: Figural Drawing Scale for Body Image Assessment (Body dissatisfaction).

The application of technologies related to the *Metaverse*, more specifically using VR in the case of these 10 studies, fulfilled diverse aims: in the first study [116], four sessions of VR exposure were conducted to evaluate the effectiveness of VR in reducing fear of weight gain and improving body satisfaction in people suffering from *Anorexia nervosa* (AN). The second study [117] focused on providing preliminary evidence on VR body exposure therapy in patients with AN, with a emphasis on fear of weight gain and body dissatisfaction. Five sessions were conducted with increasing BMI avatars. The third study [118] investigated the effects of *Full Body Illusion* on reducing body size overestimation in patients with AN. The fourth study [119] focused on whether VR could help unlock body image memory in patients with AN by combining autobiographical memories and body change techniques.

The fifth study [120] examined the effectiveness of VR-supported CBT in the treatment of body image in patients with AN and *Bulimia nervosa* (BN). The sixth study [121] investigated the effectiveness of VR-supported CBT in patients with AN. In the seventh study [122], the ability of VR to change body size perception in patients with AN was explored. The eighth study [123] was divided into two groups (experimental and control) and two *Full Body Illusion* conditions (synchronous and asynchronous) to assess size estimation of different body parts before and after VR exposure. The ninth study [124] consisted of four sessions of VR exposure where participants adopted egocentric and allocentric perspectives to induce the *Full Body Illusion*. Finally, the tenth study [125] evaluated the ability of a VR-based *Full Body Illusion* to improve body image perception and estimation of different body parts in patients with AN.

In terms of outcomes, VR body exposure therapy reduced AN symptoms, most notably the fear of weight gain, but also improved body satisfaction [117,118,121]. Likewise, VR also improved the perception of body size and/or of diverse emotionally relevant body parts for the patients [119,124,125]. Additionally, many of these positive effects were seen to persist over time, even 5 months to one year later [117,121]. Other symptoms, such as body

anxiety and the desire to lose weight, also displayed a significant reduction [117,122,123]. Finally, while Provenzano et al. [124] did not see improvements in body image distortion, they observed that this could be linked more to cognitive–emotional aspects than to visual perception. However, while Behrens et al. [116] managed to evoke and improve fear of weight gain by means of VR body exposure, this intervention did not improve body image distortion.

### 3.5. Secondary Outcomes

Regarding points 12 (main advantages and disadvantages), and 13 (student's satisfaction), the subsequent findings were retrieved (Table 4): all studies (*n* = 10; 100%) pointed out several strengths and weakness of using VR with patients suffering from AN or BN (Table 4).

**Table 4.** Secondary outcomes (*n* = 10).

| Study | Main Advantages and Disadvantages | Patient's Satisfaction |
|---|---|---|
| Behrens et al. [116] | Yes | Yes |
| Porras-Garcia et al. [117] | Yes | Yes |
| Porras-Garcia et al. [118] | Yes | Not provided |
| Keizer et al. [119] | Yes | Not provided |
| Malighetti et al. [120] | Yes | Not provided |
| Marco et al. [121] | Yes | Not provided |
| Porras-Garcia et al. [122] | Yes | Not provided |
| Porras-Garcia et al. [123] | Yes | Not provided |
| Provenzano et al. [124] | Yes | Not provided |
| Serino et al. [125] | Yes | Not provided |

Among the advantages mentioned, VR represents a technology which successfully decreased body image disturbance [117,118,120,121,123,125], fear of weight gain [116–118,123], body anxiety [117,123], ED symptomatology [117,118,121,123], and the misestimation of body size [119]. VR was also helpful in achieving a faster weight restoration than the control group [122], an increase in BMI [118,123], in some cases resulting in a healthy BMI [117], and a desired body that was closer to a normal BMI [120]. Additionally, the possibility of developing simulations with avatars helped some patients induce a strong embodiment of differently sized avatars [124]. Some studies pointed out that the treatment of an ED with the use of VR was more effective than usual CBT [118,121], and that VR technology allows patients to be exposed to their real body weight as well as a virtual representation of their own body, with more weight gain [118]; due to the modification of the size or shape of the virtual body which VR allows, to better the patient's body image disturbance.

Some of the included studies have referred to several disadvantages after using technologies related to the *Metaverse*. Porras-García et al. [117] noted that VR can be costly because it requires specialized hardware, thus limiting its accessibility for certain patients. The potential discomfort [119] caused by the increased anxiety that certain patients feel during the virtual immersive experience is also mentioned, and other authors also recommend the need to guarantee safety during interventions, in addition to minimal training and experience in application [121]. On the other hand, some patients have asserted quite the contrary, stating that the avatar and the virtual room lack realism [117], and that the avatar does not resemble them in appearance or age [118]. Specifically, in the study conducted by Porras-García et al. [118], adult patients considered the avatars to be "too young", which could explain their lower levels of full body illusion (FBI), compared to the younger patients. Some of the VR technologies used to create avatars cannot be personalized exactly how the patient would like (the outfit, tone of skin, hairstyle, etc.), which can limit or influence the effectiveness of the treatment [118]. Finally, a further disadvantage in the use of VR was that some patients had a tendency to experience dizziness [53,69,77].

Only two studies (20%) provided information regarding the patient's satisfaction after using VR [116,117]. In the first study [116], 95% of the patients said they would recommend VR exposure to other patients and in the second study [117] the patient noted that she felt "progressively more relaxed during the session and noticed the illuminated body part less" which helped her concentrate on other body parts of the avatar.

## 4. Discussion

Although previous studies have addressed the usefulness of different technologies to improve body image disturbance, to the best of our knowledge, this is the first systematic review to address the use of metaverse-related technology and its positive impact in patients suffering from AN or BN.

### 4.1. Metaverse-Related Technologies Usefulness to Improve Body Image Disturbance for Patients Diagnosed with Anorexia or Bulimia Nervosa

Metaverse-related technologies seem to be useful in improving body image disturbance (BID) for patients diagnosed with anorexia or *Bulimia nervosa* [117–121,123,125]. Specifically, VR technology is what is being used in these studies to treat BID. With the use of VR technology as a tool for patients with AN or BN, patients were able to successfully reduce their body image disturbance [117,118,120,121,123,125], their fear of weight gain [116,118,123], body anxiety [117,123] and ED symptomatology [117,118,121,123]. A number of the studies reviewed supported the idea that this kind of technology can be beneficial [117–123,125]. Furthermore, exposure to VR settings specifically designed to address fears related to weight gain (and body image distortion) may offer a valuable complement to traditional CBT [118,123]. These findings seem to be aligned with studies that were excluded [49,63]. Similarly, numerous studies validate the ability of VR to improve body image distortion in the non-clinical AN and/or BN population [48,62]. However, only some of the studies showed a statistically significant improvement in any of the body image distortion components [118,121,122], specifically, those that combined CBT with VR body exposure. This seems to indicate that VR is more effective in improving body image distortion when it is combined with a first-line treatment such as CBT than when applied alone. In fact, these results concur with other ground-breaking studies on patients with AN [139], those with diverse ED [140,141], and also those with binge-eating disorders and obesity [142].

One possible explanation for these findings could be the advantage that VR has when compared solely with other treatments such as CBT in treating body image distortion. Its main advantage is the ability to digitally manipulate the patient's weight and body image and to help the patient embody that exposure. This is not possible through traditional exposure using CBT. As such, VR technology may represent a promising tool to use in addition to face to face clinical intervention. However, another possible reason for the widespread absence of a significant improvement in body image distortion could be the reduced sample size, as pointed out by certain authors [116], with the most representative ones being studies of a single case [117,123,125]. Furthermore, certain challenges were noted, such as negative initial reactions and the need to conduct broader clinical trials to confirm the effectiveness of VR in treating body image distortion [117,122,123,125]. Maybe, the recent technological development which improves human quality interaction as MX or Metaverse may enhance patient user experience, avoiding those negative initial reactions previously mentioned.

### 4.2. Metaverse-Related Technologies Advantages to Improve Body Image Disturbance for Patients Diagnosed with Anorexia or Bulimia Nervosa

Metaverse-related technologies present various advantages for the improvement in body image disturbance in patients suffering from AN or BN. The main advantage of using VR to treat body image disturbance (BID) is that in many cases it not only improves one's perception of their own body, if not, it seems to be more effective than the usual CBT [118,121]. VR allows the manipulation and the representation of the desired body

image that one has of themselves, their real body weight, and even their body with more weight gain [118]. All of this allows patients to work on their body image disturbance through a more immersed experience, which probably results in a better understanding of their own body than what is typically achieved after only CBT techniques. VR allows patients to model and represent their ideas of their body image, which in hand allows them to deal with their body distortions; furthermore it allows patients to face their fears in a controlled environment before dealing with these issues in the real world; and lastly VR serves as an objective judge, which helps patients accept the information given to them about their body image with less resistance [36]. VR also allows good internal validity, which is obtained by the control of various variables [44]. This allows therapists to present different situations or create modifications depending on what is needed for the patient. For example, in many of the studies [116–125], they created larger or smaller bodies, using VR technology, so that the patient would experience what it was like to have said body. This immersive experience is one that a patient would not have been able to experience without the use of VR technologies. Furthermore, VR allows researchers to gather an abundance of information and responses emitted by the patient, such as eye movements or even facial gestures [143]. This helps to know which body parts create more anxiety or displeasure, as well as which body parts the patient focuses more on or directly avoids. Finally, it is very flexible and adaptive to the specific needs of each patient [143]. Regarding all the previously mentioned advantages, it should be noted that young people (ages in which BN and specially AN are more prevalent) are familiar with this kind of technology because many of them are using it by playing videogames or shopping in the Metaverse, among others. As they are familiar with Metaverse-related technology, it could be easier to introduce it in a clinical setting, in contrast to older patients.

On the other hand, the use of VR can present some disadvantages in clinical settings, such as, simulator sickness or visual fatigue [144]. Although, this is usually because there is a need for VR training and practice, before its use in clinical settings [102]. Also, depending on the VR technology, not all avatars are able to be created exactly how the patient would like (meaning there is a lack of personalization) [122], which is sometimes due to the cost of the equipment [117] which can negatively influence the effectiveness of the treatment. A main disadvantage, is the lack of standardized protocols to properly train health workers, in order to achieve empirical evidence of their effectiveness. Similarly, the high cost of this technology pointed out that financial support could be helpful to be able to acquire and use Metaverse-related technology in public hospitals, in which many BN and AN patients could take advantage of this kind of technology-based protocols.

### 4.3. Patient's Satisfaction after Using Metaverse-Related Technologies to Improve Body Image Disturbance

Only two studies explicitly mentioned patients' satisfaction after using VR, which was positive, in both cases. In the first study, 95% of the patients would recommend VR exposure to others [116] and in the second study, the patient stated that she felt more relaxed after using VR.

Although many of the studies didn't mention or measure patients' satisfaction, all the studies that were analyzed resulted in patients getting better, meaning they experienced positive outcomes. There is a strong positive relationship between the clinical outcome and a patient's satisfaction with the treatment [145]. Therefore, if the outcome of the treatment is positive, the more likely the patient is to be satisfied with said treatment. Accordingly, after examining all the approved studies one could infer, because of the positive outcomes, that the patients, in general, were satisfied without having explicitly measured their satisfaction of the use of VR to improve their body image disturbance. Another argument that supports this inference is the fact that a lower dropout rate was observed during the treatment process in the study by Marco et al. [121] than in the control group that received only CBT, which could indicate greater adherence to VR treatment. This finding is also shared by other previous studies [141].

However, while just 2/10 articles evaluated the patients' satisfaction with VR treatment, other studies have demonstrated positive acceptance outcomes of this therapy. Matsangidou et al. [146] applied a Multi-User Virtual Reality System (MUVR) on patients without a clinical diagnosis of ED but with weight and body figure distortion. The participants stated that they were comfortable with the VR system and that it was easy to use and interact with. Another positive feature provided by Matsangidou et al. [146] was the detailed customization of the participants' avatar, which generated greater immersion. Some participants in the included studies had complained precisely about the lack of resemblance with their virtual avatars [117,118], which could entail less adherence and satisfaction with the VR therapy. Thus, an important variable that improves VR body image therapy is precisely increasing the degree of realism of the avatars with a view to improving, in turn, the satisfaction of the patients undergoing this therapy. It is remarkable that patient's satisfaction level could be modified based on the resemblance with their virtual avatars provided. And this is one main advantage that Metaverse-related technology could achieve in a short time, compared to traditional VR technology.

## 5. Conclusions

Based on the outlined objectives and the findings derived from this systematic review, the following conclusions can be drawn:

(1) There is enough empirical evidence that supports the proposition that Metaverse-related technology is a promising tool to be used with patients diagnosed with AN and BN, in order to improve Body image distortion.

(2) Considering the different Metaverse-related technology developed with AN and BN patients, it is crucial to address both the advantages and disadvantages highlighted earlier in order to successfully develop Metaverse strategies to improve body image disturbance.

(3) Insufficient evidence exists in the literature regarding AN and BN patients' satisfaction level, after using Metaverse-related technology to improve body image distortion.

### 5.1. Clinical and Researcher Implications

Future randomized controlled trials are imperative to address the remaining questions such as whether Metaverse-related technology depends on the type of TCA involved (i.e., Obesity of Binge eating disorder), or if some specific modalities of Metaverse-related strategies may lead to greater improvements, engagement, and skills acquisition in patients.

### 5.2. Limitations

The main limitations of this study include the great methodological heterogeneity of the papers included. This hinders the uniformity of the results, as well as their generalizability. Moreover, most of the included studies were executed in developed countries, rather than developing countries. Therefore, in order to produce meta-analyses that provide conclusive results on the effectiveness of this approach, greater homogeneity in the targeted patients as well as in the methodology used, dates and countries in which were developed would be desirable.

Likewise, the devices used for Metaverse-related technology are constantly evolving and experiencing rapid advancements, making it imperative to update and renew the advantages and disadvantages, included in many studies, so that they are in compliance with the latest technological developments. Furthermore, the high cost of this technology, as well as the lack of standardized and accessible training for professionals who are interested in using Metaverse-related technology, represent serious barriers if one of the main goal is to achieve adequate training for its correct and effective use [147].

**Supplementary Materials:** The following are available online at https://www.mdpi.com/article/10.3390/electronics12224580/s1, Multimedia Supplementary File S1: Study protocol. Multimedia Supplementary File S2: Quality assessment of included studies (MMAT). Multimedia Supplementary File S3: Reasons for study exclusion. Refs. [148–150] are mentioned in Supplementary Materials.

**Author Contributions:** E.R. led the conception and design of the study, screening of included studies, data analysis and interpretation and wrote the first draft of the manuscript. M.P. and A.P.-R. were responsible for data extraction M.P., A.P.-R., M.P.E.-R., M.G.-M. and E.R. substantially contributed to analysis, data interpretation, and revised the work critically. All authors gave the final approval of the version to be published and agreed to be accountable for all aspects of the work by ensuring that questions related to the accuracy or integrity of any part of the work are appropriately investigated and resolved. All authors have read and agreed to the published version of the manuscript.

**Funding:** This research received no external funding.

**Data Availability Statement:** The data can be shared up on request.

**Conflicts of Interest:** The authors declare no conflict of interest.

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
