# Peer review of "Addressing Body Image Disturbance through Metaverse-Related Technologies: A Systematic Review"

_electronics, doi:10.3390/electronics12224580_

Round 1

Reviewer 1 Report

Comments and Suggestions for Authors

The manuscript entitled, “Addressing Body Image Disturbance through Metaverse-2 Related Technologies: A Systematic Review”, presents a systematic review focusing on the efficacy of Metaverse-related technologies in reducing body image disturbance related to eating disorders. The review is designed to answer certain questions highlighted in the article, following the PRISMA statements.

As a reviewer, below are my comments and suggestions regarding the presentation and content of the manuscript:

1.       The authors mentioned, “It was observed that the prevalence for AN in woman was between 0.67-1.2% while in men it was 0.1%. Whereas for BN, the prevalence in woman was 0.62%.” Here, please mention the prevalence of BN in men.

2.       The information presented in the column “Training Using Metaverse-Related Technologies” of Table 3 could be discussed in the main text body. This will optimize the size of the table and will provide better clarity to the readers.

3.       Although an attempt to answer three broad questions in the article has been made, it would be great to discuss some of the questions highlighted in section 5.1 in the manuscript based on the review.  

Minor comments: There are some minor typos which need to be addressed. For instance, the author mentioned, “It was observed that the prevalence for AN in woman was between 0.67-1.2% while in men it was 0.1%.” Here, it should be “women”. Similarly, there are some instances which need a thought revision of the manuscript to avoid any typos/grammatical errors.

Comments on the Quality of English Language

The text can be improved to avoid any grammatical errors/typos.

Author Response

Dear reviewer.

We do really appreciate your kind report about our manuscript.

Please, find attached the point-by-point response to each comment suggested.

Kind regards,

Authors

Reviewer 2 Report

Comments and Suggestions for Authors

This systematic review paper have a clear focus on examining the efficacy of Metaverse-related technologies in reducing body image disturbance related to eating disorders like Anorexia nervosa and Bulimia nervosa. It would be interesting to see if the review also explores the potential limitations or challenges of using Metaverse-related technologies in this context. However, due to insufficient systematic clarified, the conclusion looks not enough for readers. Here are some questions and suggestions:

1.It needs to be explained why it started from 2023, without including early research.

2.Does the search term involve approximate words? Search formula is suggested to be listed clearly.

3.The use of Cohen kappa coefficient recommend to add the reference.

4.The list way in table 3 and 4 looks provided limited information. Review paper is not only a paper list, but also summary and comparison for valuable information. It would be helpful to see more details on the specific outcomes or measures that will be analyzed and visualized them by a simple way to support the discussion.

5.Focusing on trends and insights into challenges in discussions and conclusions would be more valuable and necessary.

Comments on the Quality of English Language

Minor editing of English language required

Author Response

(The authors gave the same response as above.)

Reviewer 3 Report

Comments and Suggestions for Authors

I will send you my report as an attachment. Thank you for inviting me to review this interesting study.

Author Response

(The authors gave the same response as above.)

Round 2

Reviewer 2 Report

Comments and Suggestions for Authors

I agree to publish this paper.

Comments on the Quality of English Language

 The English language will be fine after minor edits.